# AKVANO^®^: A Novel Lipid Formulation System for Topical Drug Delivery—In Vitro Studies

**DOI:** 10.3390/pharmaceutics14040794

**Published:** 2022-04-05

**Authors:** Jan Holmbäck, Vibhu Rinwa, Tobias Halthur, Puneet Rinwa, Anders Carlsson, Bengt Herslöf

**Affiliations:** 1Lipidor AB, Svärdvägen 13, SE-182 33 Danderyd, Sweden; vibhu.rinwa@lipidor.se (V.R.); puneet.rinwa@lipidor.se (P.R.); 2Department of Materials and Environmental Chemistry, Stockholm University, Svante Arrhenius väg 16C, SE-106 91 Stockholm, Sweden; 3CR Competence AB, Naturvetarvägen 14, SE-223 62 Lund, Sweden; tobias@crcom.se; 4Biofilms—Research Center for Biointerfaces, Department of Biomedical Science, Faculty of Health and Society, Malmö University, Per Albin Hanssons väg 35, SE-214 32 Malmö, Sweden; 5MediGelium AB, Hornsbergs strand 49, SE-112 16 Stockholm, Sweden; medigelium@telia.com; 6Lipidea AB, Brunbärsvägen 2, SE-114 21 Stockholm, Sweden; bengt.herslof@lipidor.se

**Keywords:** topical drug delivery, phospholipids, physicochemical methods, atomic force microscopy, antibacterial, antiviral, antifungal, in vitro skin irritation, in vitro eye irritation, in vitro skin permeation

## Abstract

A novel formulation technology called AKVANO^®^ has been developed with the aim to provide a tuneable and versatile drug delivery system for topical administration. The vehicle is based on a water-free lipid formulation where selected lipids, mainly phospholipids rich in phosphatidylcholine, are dissolved in a volatile solvent, such as ethanol. With the aim of describing the basic properties of the system, the following physicochemical methods were used: viscometry, dynamic light scattering, NMR diffusometry, and atomic force microscopy. AKVANO formulations are non-viscous, with virtually no or very minute aggregates formed, and when applied to the skin, e.g., by spraying, a thin film consisting of lipid bilayer structures is formed. Standardized in vitro microbiological and irritation tests show that AKVANO formulations meet criteria for antibacterial, antifungal, and antiviral activities and, at the same time, are being investigated as a non-irritant to the skin and eye. The ethanol content in AKVANO facilitates incorporation of many active pharmaceutical ingredients (>80 successfully tested) and the phospholipids seem to act as a solubilizer in the formulation. In vitro skin permeation experiments using Strat-M^®^ membranes have shown that AKVANO formulations can be designed to alter the penetration of active ingredients by changing the lipid composition.

## 1. Introduction

Topically administered pharmaceutical and cosmetic products could be designed to exert their action locally, regionally, or systemically. Within each of these categories, the target site could be more or less defined. Products for local action could, for instance, target the skin surface, viable epidermis, dermis, or skin appendages such as hair follicles. The target could also be nerves or muscles or to achieve absorption by the circulatory system (transdermal delivery). Each of these delivery routes demands a well-designed formulation system, which also must take into consideration aspects such as solubility and stability of the active ingredient.

This is the first publication describing a novel topical delivery system called AKVANO^®^ (an abbreviation for “water-free”), which is aimed to be tuneable to target the desired site, while minimizing local irritation and providing a convenient mode of application for the user. The AKVANO technology is a formulation platform for topical delivery of various pharmaceutical ingredients but has also found use in consumer health care products and topical products for animal care. It is based on water-free lipid formulations where phospholipids and elective active ingredients are dissolved in volatile solvents.

Membrane-forming lipids like phospholipids display a rich phase behaviour in water. These lipids do not dissolve in water. Instead, due to their amphiphilicity, they interact with water in different ways depending on their chemical structure, concentration, and temperature [1]. Different liquid crystalline structures may be formed, such as lamellar bilayers and reversed hexagonal phases [2]. These phases are formed at a relatively low concentration in water and are highly viscous. Upon further dilution with water, colloidal dispersions are formed, most notably the vesicles and liposomes are formed from lamellar phases. Dispersions are thermodynamically unstable but may be kinetically stabilised to slow down the phase separation process [3]. Dispersions are normally opaque or cloudy due to the large sizes of the colloids which give rise to significant light scattering [2].

The AKVANO drug delivery system is based on the idea of utilizing a liquid lipid system without any interactions between lipids and water. The system therefore consists of lipids dissolved in a water-free solvent system. Suitable solvents are volatile alcohols and silicone oils. Such solutions have a low viscosity even with a relatively high concentration of lipids, e.g., 25% by weight or more, and good incorporation ability for drugs and excipients. AKVANO formulations can be easily applied to the skin, e.g., by spraying. Due to the evaporation of the volatile solvent, it results in a thin lipid film containing the active ingredient.

The principal composition of the AKVANO system is a membrane-forming lipid, i.e., a double-chain lipid in a volatile solvent system. Since skin lipids are organised in bilayer structures [4], lipids that are able to form bilayers, such as phosphatidylcholine, are considered as especially useful. By adding other lipid types, e.g., single-chain lipids, the composition is variable and can be adapted to specific applications, e.g., to a specific active pharmaceutical ingredient or to a specific function or layer of the skin (thus providing protection or enhancing penetration). The AKVANO solvent is typically a short-chain alcohol such as ethanol. In contrast to water ethanol, it can dissolve the lipids, forming an unorganised (isotropic) solution in which the lipid molecules are apparently molecularly dispersed. The absence of organisation, i.e., no significant presence of ordered self-assembled aggregates, makes the solution low-viscous, with a viscosity close to that of the neat solvent. A solution is also thermodynamically stable. Alcohols are not only good solvents for lipids, but also for a great number of active ingredients, which is an additional advantage. The alcohol can be mixed with other volatile solvents such as siloxanes, both cyclic (e.g., cyclomethicone) and short straight-chain siloxanes (e.g., hexamethyldisiloxane).

Several AKVANO compositions have already been described in a series of published patents (list attached). To date, more than 80 active ingredients have been successfully incorporated into AKVANO that are forming clear and stable low-viscous solutions, and for some active pharmaceutical ingredients (APIs), the phospholipids act as a solubilisation agent. In this article, the results from several in vitro tests are described. The solution properties of some AKVANO compositions have been characterised by viscometry, dynamic light scattering, and NMR diffusometry, comparing two types of lipids (one membrane forming and one more prone to form reversed hexagonal phases). To study the behaviour of AKVANO formulations after application and subsequent evaporation of the volatile solvent, the properties of the films obtained for those formulations have also been studied by atomic force microscopy. Additionally, bactericidal, fungicidal, and virucidal properties of AKVANO formulations and results from in vitro skin and eye irritation tests are reported. Finally, in vitro permeation results for AKVANO formulations of ketoprofen, diclofenac diethylamine, and diclofenac sodium using artificial membranes are presented in this article.

## 2. Materials and Methods

### 2.1. Composition of AKVANO

The composition of AKVANO has been developed with the aim of integrating lipid soluble ingredients into the polar lipids while achieving a clear appearance and a low viscosity of the resulting cutaneous solution. The major portion of the total content of this solution (up to 80% or more) rapidly evaporates within minutes after application to skin, resulting in formation of a polar lipid film containing the active pharmaceutical ingredient. The principal pharmaceutical excipients in AKVANO are described below.

#### 2.1.1. Lipids

The polar lipid component is normally based on phospholipids from soybean or other plant material. In addition, single chain lipids such as monoglycerides, isopropyl myristate (IPM), or other fatty acid alcohol esters can be added. The total concentration of lipids is normally in the range of 5% to 25% by weight.

#### 2.1.2. Alcohol

The preferred alcohol is ethanol and, normally, an anhydrous quality is used to avoid unnecessary addition of water. A typical concentration of alcohol is from 20% up to as high as 95% by weight. Ethanol is an efficient preservative for AKVANO and, accordingly, no additional preservative is needed to be added to the formulation. For non-pharmaceutical products, and depending on national legislation, ethanol may have to be denatured. A denaturing system based on different short chain alcohols such as 2-propanol and tert-butanol is usually preferred.

#### 2.1.3. Keratolytic Agents

In some applications, as is the case with other topical dosage forms, addition of keratolytic agents can be beneficial. Examples of keratolytic agents suitable to be used with AKVANO are α- and β-hydroxy acids, such as glycolic acid, lactic acid, malic acid, salicylic acid, and their salts. Another suitable keratolytic substance is urea.

#### 2.1.4. Silicone Oil

Part of the volatile solvent system can consist of volatile silicone oil such as a cyclomethicone, or specifically decamethylcyclopentasiloxane (also denoted as cyclomethicone D5). It is a fully methylated cyclic siloxane containing five repeating units of the formula ((CH_3_)_2_SiO–). It is used in a concentration of up to 60% by weight. Despite a rather high boiling point, cyclomethicone D5 has a high volatility due to its low enthalpy of vaporization. This gives a ‘dry’ feeling when applied to the skin (no cooling).

#### 2.1.5. Active Agents

As mentioned earlier, many active substances have been tested in AKVANO and some are listed in the patent applications. The successfully tested substances in our lab typically have a positive logK_ow_ and a molecular weight below 1000 g/mol. Examples of drugs that have been tested are anti-psoriatic, anti-acne, anti-eczema, antimicrobial, anti-inflammatory, antifungal, and wound healing agents, as well as local anaesthetics and non-steroidal anti-inflammatory drugs (NSAIDs).

#### 2.1.6. Additional Components

Depending on the desired properties of the final formulation, other substances such as fragrances, essential oils, and thickeners can be added.

### 2.2. Preparation of AKVANO

In brief, a general procedure to prepare an AKVANO formulation can be described as follows. Phospholipids with additional lipids are weighed and mixed with part of the ethanol until a clear solution is obtained. In a separate vial, active pharmaceutical or cosmetic ingredient(s) are weighed and dissolved in the remaining ethanol. The mixture is stirred using a magnetic stirrer or impeller until a clear solution is obtained and, finally, active solution and, optionally, a silicone oil (such as cyclomethicone) is added. Depending on the compatibility, keratolytic agents and other optional ingredients can be added to either of the before mentioned solutions prior to mixing them. The procedure may be further adjusted depending on the requirements of the specific formulation, and especially when the batch size is scaled up.

The composition of different formulations used in this paper are listed in Table 1. Phospholipids used are either the synthetic dimyristoylphosphatidylcholine (DMPC), dioleoylphosphatidylethanolamine (DOPE), natural soybean lecithin (Soy-Lec), soybean phospholipids (Soy-PL), or soybean phosphatidylcholine (Soy-PC). All phospholipids were obtained from Lipoid AG (Steinhausen, Switzerland). Monoglycerides used were medium chain monoglycerides (MCM) from Abitec Corp. (Columbus, OH, USA). Isopropyl myristate (IPM), oleic acid, α-tocopherol, citric acid, tert-butanol, and ketoprofen were from Sigma–Aldrich (Darmstadt, Germany). Absolute ethanol (EtOH) was from VWR (Stockholm, Sweden). Cyclomethicone was supplied by DuPont de Nemours, Inc. (Wilmington, DE, USA). Calcipotriol was supplied by Cerbios–Pharma SA (Barbengo-Lugano, Switzerland). Diclofenac diethylamine and diclofenac sodium were from Dipharma, (Mereto di Tomba, Italy) and betamethasone dipropionate from Farmabios SpA (Pavia, Italy). In addition, 2-propanol and HPLC solvents were from Rathburn Chemicals (Walkerburn, UK).

### 2.3. Physicochemical Characterization of AKVANO Formulations

These characterizations were carried out to obtain a better understanding of the extent of lipid aggregation and influence of water on AKVANO formulations based on a solvent mixture of ethanol (EtOH) and cyclomethicone D5. The formulations were based on two different types of lipids: DMPC, which is known to form bilayer membrane structures, or DOPE, which is more prone to form reversed hexagonal structures and the corresponding formulations with added water (~1.5%). The formulations were characterized using viscometry, dynamic light scattering (DLS), and nuclear magnetic resonance (NMR) diffusometry.

#### 2.3.1. Viscometry

Viscosity measurements were performed on formulations with and without the addition of water, as well as on a sample without lipids on a Lovis 2000M Microviscometer (Anton Paar, Graz, Austria) at 25 °C. A thin capillary (1.59 mm in diameter) was filled with the sample liquid using a syringe. A small steel ball (1.5 mm in diameter) was then introduced, and the capillary was sealed off with a lid making sure that no air bubbles were present. During measurements, the capillary was turned at an angle of 80° and the time taken by the steel ball to descend through the capillary was measured with an accuracy of 0.05%. The capillary was then turned to 80° in the opposite direction so that the ball falls back and the time for the return was measured. Four consecutive measurements were made on each sample to ensure good accuracy.

#### 2.3.2. Dynamic Light Scattering (DLS)

DLS experiments were performed on a Malvern Zetasizer Nano ZS (Malvern Instruments, Malvern, UK) at 25 °C. The formulation samples were filtered with Minisart SRP 25 PTFE membrane (0.45 µm) (Sartorius Göettingen, Germany) before measurements to remove larger particles, such as dust. The samples were equilibrated for 5 min before the experiments and three measurements were performed on each sample. The viscosity was set to the value determined for the pure silicon oil/EtOH (74.7/25.3%) solvent mixture, i.e., 2.37 mPa·s, and a refractive index of 1.48 was used for the dispersed phase and all samples were analysed in triplicates.

#### 2.3.3. NMR Diffusometry

The experiments were performed on a Bruker AVII-200 spectrometer equipped with a Bruker DIFF-25 probe and a Bruker GREAT 1/40 gradient amplifier (Bruker, Billerica, MA, USA). The temperature control was calibrated using a thermocouple immersed in an NMR tube to measure the actual temperatures at the position of the sample. The self-diffusion coefficients were determined using the pulsed gradient stimulated echo method [5] with a pulsed-field gradient width of 1 ms, a diffusion time (∆) of 20 ms, and an acquisition time of 1 s. The gradient strength was linearly ramped in a range selected to obtain an appropriate decay of the spin echo in the respective experiments. For each spectrum, eight scans were recorded with a repetition delay of 1 s. Each experiment was preceded by four dummy scans.

### 2.4. Characterization of the Film Formed after Evaporation by Atomic Force Microscopy (AFM)

This study was carried out using a XE-100 (Park Instruments, Suwon, Korea) to investigate whether the lipids dissolved in silicone oil and ethanol forms lipid bilayers when dried on a hydrophilic substrate. This was conducted by letting a small drop of formulation dry on a hydrophilic silica surface and then measuring the layer thickness with AFM. The formulations were first diluted 100 times in the ethanol (22.2%)/cyclomethicone (77.8%) solution to reduce the overall thickness of the coating formed, and thus improving the chances of finding single bilayers deposited on the substrate. A drop of diluted formulation was applied to a clean silica substrate (boiled in acid and base just prior to use) inside a laminar air flow (LAF) bench, and the substrates were then leaned at a 45° angle, thereby creating a thickness gradient, with the thinnest coating at the upper part of the substrate. The substrates were left in the LAF bench to dry overnight. AFM images were produced by scanning the surfaces in intermittent contact mode in air (using PPP-NCHR cantilever from Park Systems, Suwon, Korea) at scan speeds 0.7–1 Hz, while recoding the topography, amplitude, and phase signal. Images were evaluated in the XEI Park instruments software (Park Systems, Suwon, Korea), where profile lines were drawn in selected locations to measure the step-height for the lipid bilayer structures.

### 2.5. Antimicrobial Tests

#### 2.5.1. Antibacterial Activity

The test was carried out by QACS Ltd. Laboratory (Athens, Greece) as per the European Standard test method EN 1500:2013 [6]. The method was specified for verifying hygienic hand rub where the test product (PP), when rubbed onto artificially contaminated hands of volunteers, should reduce the release of transient flora. The live test organism (*Escherichia coli* K12 NCTC 10538) was applied and recovered to obtain a baseline count. The test product (PP)/reference product (RP) is later applied to terminate the effect of any residual disinfectant before recovering any surviving test organisms in sampling broth containing neutralizers. AKVANO skin disinfectant formulations were used as formulation products and 2-propanol, 60% in water (*v*/*v*) was used as a reference. The organisms were enumerated, counts transposed to the log_10_ (log) system, and the difference between the numbers recovered from the AKVANO or reference formulations, and baseline counts were established and statistically analysed for any significance. The larger the difference is between the two counts, the less effective is the product. Each of the volunteers repeated the procedure for the reference first and the product to be evaluated after, and then for the product first and the reference after. AKVANO foot spray formulation was also tested for antibacterial activity by Lab-test laboratorium S.C. (Katowice, Poland) according to European Standard [7]. In brief, the test method is dilution–neutralization with neutralizer D/E broth, at clean conditions (0.3 g/L bovine albumin), contact time 30 s, and test temperature 20.0 °C ± 0.6 °C, diluted in distilled water against *Pseudomonas aeruginosa* (ATCC 15442), *Staphylococcus aureus* (ATCC 6538), *Enterococcus hirae* (ATCC 10541), and *Escherichia coli* K12 (NCTC 10538).

#### 2.5.2. Antifungal Activity

AKVANO formulation was tested for antifungal activity by Lab-test laboratorium S.C. (Katowice, Poland) according to European Standard [8]. This European Standard specifies a test method and the minimum requirements for fungicidal activity of chemical disinfectant and antiseptic products. AKV014 was evaluated at clean conditions with interfering substance 0.3 g/L bovine albumin at a contact time of 30 s and test temperature of 20.0 ± 0.6 °C diluted in distilled water. Formulation was tested at concentrations of 10–97% *v*/*v*. The incubation time was 48 h using pour plate method at 29.5–30.5 °C. At the end of this contact time, an aliquot was taken and the fungicidal action against microbial strain *Candida albicans* (ATCC 10231) in this portion was immediately neutralized or suppressed by a validated method (dilution–neutralization). The numbers of surviving fungi in each sample were determined and the reduction was calculated.

#### 2.5.3. Antiviral Activity

The test was carried out by QACS Ltd. Laboratory (Athens, Greece) as per the European Standard test method [9]. The antiviral activity of AKVANO formulations (intended for use as skin disinfectants) was tested against three virus strains: Adenovirus type 5, Poliovirus type 1, and Murine norovirus. A 97% dilution of the AKVANO product was added to a test suspension of titrated viruses in bovine serum albumin solutions of 0.3 g/L (clean conditions). The mixtures were maintained at 20 °C for 60 s. At the end of contact time, an aliquot was taken and the virucidal activity was suppressed by dilutions in ice-cold maintenance medium. The dilutions were then inoculated onto cell monolayers in 96-well culture plates for the titration of the remaining viruses. The titres of the viruses expressed in the Tissue Culture Infectious Dose (TCID50) values, after 5-days of incubation, were determined and expressed in a log scale. Reduction in the virus infectivity was calculated from the differences of the log virus titres before (control) and after treatment with the AKVANO product. According to the EN 14476 standard, a product has antiviral activity when the reduction of the virus is at least four log units.

### 2.6. In Vitro Tests for Irritation Potential

#### 2.6.1. In Vitro Skin Irritation Test

This test was carried out by QACS Ltd. Laboratory (Athens, Greece) according to the Organisation for Economic Co-operation and Development (OECD) Guideline No. 439 [10] and using the protocol In Vitro EpiDerm™ Skin Irritation Test [11]. Skin irritation refers to the generation of reversible damage to the skin following the exposure of the chemical to be evaluated, for up to 4 h [10]. The test consisted of a topical exposure of AKVANO formulation (intended for use as a skin disinfectant) to a reconstructed human epidermis (RhE) model followed by a cell viability test. Cell viability was measured by dehydrogenase conversion of MTT present in cell mitochondria into a blue formazan salt that was quantitatively measured photometrically after extraction from tissue. The reduction of the average viability of three tissues exposed to chemicals in comparison to average viability of three negative controls (treated with water) was used to predict the skin irritation potential. The negative control used was DPBS without Ca^2+^ and Mg^2+^ and 5% sodium dodecyl sulphate (SDS) solution was used as a positive control.

#### 2.6.2. In Vitro Eye Irritation Test

This study was carried out by Research Institutes of Sweden AB (RISE, Gothenburg, Sweden, according to the OECD guidelines No. 492 [12] and using the protocol In Vitro EpiOcular Eye Irritation Test [13]. The eye irritation test is based on the use of a reconstructed cornea epithelial model and the relevant materials were obtained from MatTek In Vitro Life Science Laboratories (Bratislava, Slovak Republic). The epithelia models are topically exposed to the product to be evaluated and after recovery, the viability of cells is measured via metabolic activity. Yellow water-soluble MTT (3-(4,5-dimethylthiazol-2-yl)-2,5-diphenyltetrazolium bromide) is metabolically reduced in viable cells to a blue-violet insoluble formazan, and thus the number of viable cells correlates to the colour intensity determined by photometric measurements after dissolving the formazan in alcohol. For each treatment, the viability percentage relative to a negative control (cell culture water) is calculated. Positive control used was neat methyl acetate. Eye irritation is identified as the ability of the product to be evaluated to reduce the viability of the cells in the epithelial model system. Eye irritation potential of the product or formulation evaluated is predicted if the remaining relative cell viability is below 50% after exposure.

The AKVANO formulation, positive control and negative control were added to EpiOcular™ human cell construct models (MatTek In Vitro Life Science Laboratories, Bratislava, Slovak Republic) pre-treated with DPBS, Dulbecco’s Phosphate Buffered Saline (Thermo Fisher Scientific, Waltham, MA, USA) for 30 min, whereafter the tissues were thoroughly washed followed by a post-treatment immersion and then allowed to recover for 2 h. After the recovery period, MTT solution was added to the tissues which were incubated for an additional 3 h at 37 ± 1 °C in 5 ± 1% CO_2_. Following incubation, the MTT solution was removed, 2-propanol was added, and the plate with the models was shaken rapidly for at least 2 h. The solutions for tissues were homogenized and transferred to a 96-well plate for absorbance measurement at 570 nm followed by calculation of the viability of the tissues.

### 2.7. In Vitro Skin Permeation Studies

These experiments were performed using Strat-M^®^ membranes [14] from Merck (Darmstadt, Germany) to study the permeation of ketoprofen, diclofenac diethylamine, and diclofenac sodium in different AKVANO formulations and in commercially available medicaments. The diffusion cell system [15,16] consisted of an eight-channel peristaltic pump, which delivered PBS buffer pH 7.4 to flow-through diffusion cells with a cross section area of 0.5 cm^2^ placed on a stainless-steel platform which was kept at 37 °C. The receptor fluid was transported in the system through Teflon tubes (0.5 mm ID) with an approximate flow rate of 1.5 mL/h to an eight-channel fraction collector. Strat-M membranes were cut to an appropriate size and placed between the donor chamber and the receiving chamber. Approximately 5 mg of formulation was applied on top of the membranes. The opening of the donor chamber was left uncovered to allow evaporation of the volatile solvent. Receptor fluid was collected at the following time intervals: 0–2, 2–4, 4–6, 6–10, 10–14, 14–18, and 18–24 h. The concentration of active substance in the receptor fluid was analysed by RP-HPLC (Agilent Technologies Inc., Santa Clara, CA, USA) with UV detection at 240 nm for ketoprofen and 276 nm for diclofenac salts. The separation for ketoprofen was carried out on a Symmetry C8 column (150 × 3.9 mm, particle size 5 µm) from Waters (Milford, MA, USA) and for diclofenac salts (250 × 4.6 mm, particle size 5 µm) from ReproSil-Pur C8 by Dr Maisch HPLC GmbH (Ammerbuch, Germany). Ketoprofen was eluted using a flow of 1–2 mL/min with 75% A and 25% B to 100% B for 13 min, where A is methanol/water 40:60 + 0.16% triethylamine + 0.16% acetic acid and B is methanol + 0.16% triethylamine + 0.16% acetic acid, and diclofenac salts were eluted using a flow rate of 1.0 mL/min with 70% A and 30% B to 100% B in 15 min, where A is methanol/water 10:90 + 0.1% acetic acid and B is methanol + 0.1% acetic acid. The amount of active substance retained in the membranes was analysed after extraction with 1 mL of methanol overnight.

## 3. Results and Discussions

### 3.1. Physico-Chemical Characterization of Different AKVANO Formulations

Characterization of model AKVANO compositions was used to understand the significance of the choice of phospholipids to obtain the desired properties of both the bulk liquid itself and the organization after evaporation (see Section 3.2). AKVANO formulations (AKV001–AKV005) with different compositions (see Table 1) were used for characterization experiments, as follows:

#### 3.1.1. Viscometry

The presence of lipid, either DMPC or DOPE, increased the formulations’ viscosity by about 25% (Table 2), which could suggest that aggregates are formed. Furthermore, the addition of 1.5% (*w*/*w*) of water further increased the viscosity of the AKVANO formulations, which may indicate that water gives some enhancement in the aggregation.

#### 3.1.2. Dynamic Light Scattering (DLS)

Table 3 presents the average values (from triplicates) of the hydrodynamic radii obtained from the DLS measurements. When evaluating the DLS data, it is important to keep in mind that the lipid concentration in AKVANO formulations used is rather high. This ensures that if any aggregates giving rise to scattering are present, one can expect a good signal. The obtained size distributions are indeed well-defined, with a high repeatability for the triplicates, as seen by the low standard deviation. They are also monomodal with a low polydispersity index, which would suggest the presence of small aggregates (Figure 1). On the other hand, the presence of a large fraction of lipids in AKVANO formulations involves a significant risk of multiple scattering, which can give rise to overestimation of the aggregate’s dimensions. In summary, the DLS measurements suggest that the apparent average size of aggregates formed by DOPE is larger than those formed by DMPC, and that the presence of water possibly gives rise to some aggregate growth in the DOPE formulation (Table 3). Regarding the average hydrodynamic radii presented in Table 3, it can be stated that they are on the same order of magnitude as would be expected for micelles, which are generally considered to have a radius corresponding to the maximum extended length of the aggregating amphiphile (which in this case would be around 3 nm). However, since the hydrodynamic radius is generally larger than the actual physical radius (due to solvation effects), it can be stated that the data measured for all formulations are probably on the small side for micelles, and especially for the DMPC formulation exhibiting data much smaller than what would be expected for micelles.

In contrast, according to the literature, in phospholipid/ethanol systems with higher content of water, vesicular systems are formed with much larger aggregate size (ranging from one to several orders of magnitude larger) [17,18].

#### 3.1.3. NMR Diffusometry

A well-resolved diffusion coefficient for the lipid as well as for the solvent components, e.g., ethanol and cyclomethicone D5, was obtained for all the formulation samples. Diffusion coefficient of the added water was, however, only identified and analysed in the sample with DMPC. The DOPE sample with water was notably different compared to the sample without water in terms of viscosity, DLS, and diffusion coefficients of the other components. The absence of a distinct water signal in the DOPE sample with water is likely caused by the signal being buried in larger signals arising from the lipid. The possible presence of aggregates is best assessed by calculating the apparent hydrodynamic sizes corresponding to the values of *D_lipid_* (Table 4) derived from the Stokes-Einstein relation [5]:(1)$$D_{lipid}=k_{B}T/6\pi \eta R_{h}$$
where *k_B_* is the Boltzmann’s constant, *T* the absolute temperature, and *η* the viscosity of the formulation.

An important finding from these numbers is that if aggregates are formed, these are small, even smaller than indicated from the DLS data. As was mentioned above, there is a significant risk that, because of likely multiple scattering, the apparent sizes obtained from the DLS measurements are over-estimated. Since the apparent sizes obtained in the NMR experiments can be more “direct”, these can be expected to reflect the effective sizes of the diffusing entities more reliably. The apparent dimensions corresponding to the *D_lipid_* are significantly smaller than expected from micelle-like aggregates. The radius of a spherical micelle is typically close to the extended length of the monomer, which, for the investigated lipids here, is ~3 nm. In addition, the hydrodynamic radius is expected to be slightly larger than the physical radius, due to the influence from solvation. Thus, lipid self-diffusion data clearly suggest that possible aggregates formed are smaller than the typical micelles.

It can be noted that, generally, if a molecule is present both as individually dissolved monomers and residing in well-defined aggregates (such as micelles), its observed diffusion coefficient (*D_obs_*) under the condition of fast exchange between the two sites (which there is no reason to believe that one would not have) is the weighted average of the diffusion coefficients corresponding to the two sites (*D_monomer_* and *D_aggregate_*, respectively), according to Equation (2):*D_obs_* = *p_monomer_* × *D_monomer_* + (1 − *p_monomer_*) × *D_aggregate_*(2)
where *p_monomer_* is the fraction of molecules present as individually dissolved molecules. Since *D_aggregate_* is typically smaller than *D_monomer_*, it does, in relative terms, have a smaller influence on *D_obs_*, and one could, in principle, still have large aggregates present in a fraction low enough that its influence on *D_obs_* is minor. However, considering the high concentration of lipid in the investigated samples, it would be highly unlikely that, had there been a propensity for micelle formation, a major fraction of the lipid had not been residing in aggregates. As was mentioned above, *D_water_* could only be determined for sample AKV003 (DMPC + H_2_O). It is found that, although the water molecule is smaller than the ethanol molecule, *D_water_* is significantly lower than *D_EtOH_*. This finding can be taken to suggest that there is some extent of preferential binding of water to the lipid. In this context, it can be noted that, at the herein used water concentration, there are only around six water molecules per lipid molecule. It is possible that, in the presence of a larger fraction of water, the formation of more micelle-like aggregates may be induced. To obtain further understanding of the solution structure, it is valuable to take a closer look at the diffusion data for the solvent components.

Table 5 presents the diffusion coefficients of ethanol and cyclomethicone D5 in samples with lipids normalized to the corresponding values in the pure solvent mixture in the absence of lipids. By comparing the values of *D*/*D*_0_ in Table 5 to the viscosities presented in Table 2, one can find out that the relative decrease in *D* in the presence of lipid is very close to the inverse of the corresponding relative increase in viscosity. This indicates that the reduction in diffusion rate is mainly a consequence of an increase in bulk viscosity; had a significant volume fraction of large aggregates been present in the samples, one would have expected additional reduction in solvent diffusion due to obstruction effects caused by excluded volume. These findings give additional support to the notion that micelle-like aggregates are not formed in the samples. Furthermore, they suggest that there is no preferential binding of either of the solvent components to the lipid, since the diffusion coefficients for both solvents are similar within one sample.

It is difficult to tell whether some types of smaller aggregates are formed in the samples and to get an idea of their character. Considering the size of the individual lipid molecules—as is stated above, the extended length of these is ~3 nm—it is possible that there is no significant association of the lipid molecules and that they are individually dissolved in solution. It should be noted that the effective size of individually dissolved monomers is typically more strongly affected by solvation than that of molecules in aggregates, i.e., the apparent volume per lipid molecule may be notably larger for an individually dissolved molecule than for one residing in an aggregate. It is thus not possible to unambiguously separate solvated monomers from aggregates made up of a small number of lipid molecules. The NMR diffusion data do, in accordance with the DLS data, suggest that there is a stronger tendency for aggregation in the samples with DOPE than in those with DMPC.

### 3.2. Characterization of the Film Formed after Evaporation by Atomic Force Microscopy (AFM)

The film obtained after evaporation of formulation AKV006 (DMPC) showed a characteristic pattern of rounded shapes, such as flat patches or islands protruding from the underlying substrate when imaged with AFM. These rounded shapes had a step height of 4–5 nm, or sometimes multiples of that height, indicating that they are islands of bilayers and, occasionally, multiple bilayers were formed on the surface (Figure 2). The structures appeared very flat (as seen in the topography signal) and smooth (as seen in the amplitude signal, which is more sensitive to edges and smaller features). Looking at the topography and amplitude signal, the roughness appears to be very similar between the shapes as on top of them. On the other hand, the phase signal, which is sensitive to material properties such as stiffness and elasticity, clearly shows a contrast between the two locations (Figure 2c), thus indicating that the rounded shapes seen are lipid bilayers deposited on top of a clean silica substrate. This is even more obvious when looking at the few locations where an additional smaller round-shaped lipid bilayer is deposited on the top of a larger structure, since no contrast is evident when moving between the two bilayer levels, whereas the phase signal changes significantly once the bottom surface is reached (Figure 2). It can further be stated that no contrast could be seen in the phase signal when AFM scans were made on locations where a thicker coating was deposited on the substrate (Figure 3c) (further down on the substrate where more material had accumulated). AFM images taken at the “thicker” end of the surface also show much thicker stacks of lipid bilayers, as seen in Figure 3 below, although even these thick assemblies clearly indicate a layered structure where the top layers are sometimes seen to be incomplete (see green profile line in Figure 3).

The AKV007 (DOPE) sample, on the other hand, looks completely different in the AFM images, where the DOPE lipids appear to form elongated worm-like structures when dried on a hydrophilic substrate, as seen in Figure 4. This is even more pronounced when AFM scans are made at locations with thicker coatings, where long worm-like fibres are seen (Figure 5). There is no evidence of bilayer formation (as expected); instead, elongated structures possibly connected to the reversed hexagonal phase normally formed by DOPE [2] are seen. These results thus confirm that phosphatidylcholine rich lipid materials are more suitable to be used in AKVANO than materials with a high content of phosphatidylethanolamine.

### 3.3. Antimicrobial Tests

#### 3.3.1. Antibacterial Activity

For test formulations to confirm to the standard method (EN1500:2013), the mean log reduction factor obtained should not be inferior to that achieved by the specified reference product. The acceptance criteria as laid out by European standard EN1500:2013 were fulfilled with AKVANO formulations, as primarily individual log reductions were less than 3.00 and means of log prevalues for AKVANO formulations (7.12) and for reference product (7.07) were greater than 5, with absolute difference of mean differences as 0.12 (hence less than 2.00).

The performance of AKVANO formulations in the test procedure proved to be equivalent to the performance of the reference product (RP). AKVANO formulations AKV012 and AKV013 showed a log reduction of 3.9 and 4.3, respectively, while the log reduction of RP was 3.8 and 3.6 at the respective testing occasions (Figure 6). The relatively higher bactericidal effect of AKV013 could be attributed to the content of citric acid in the formulation. Accordingly, both versions of AKVANO skin disinfection spray, tested at 100% concentration, when applied for total rubbing time of 60 s (2 × 30 s) and using total quantity of 6 mL (2 × 3 mL dose) of product, conforms to the requirements of EN 1500:2013.

An AKVANO foot spray formulation, AKV014, was also tested according to EN 13727+A1:2014-02 and proved to have met the efficacy requirements in reduction of viable counts (five log units) against *Pseudomonas aeruginosa*, *Staphylococcus aureus*, *Enterococcus hirae*, and *Escherichia coli* at 80% (*v*/*v*) and at 97% (*v*/*v*).

#### 3.3.2. Antifungal Activity

According to the test procedure EN 13624, the product should demonstrate at least a four-decimal log reduction to pass the acceptance criteria. AKVANO formulation (AKV014) was tested at concentrations of 10%, 80%, and 97% (*v*/*v*) and the reduction factor of viable counts (R) was found to be active (reduction > 4.48 log units) against *Candida albicans* ATCC 10231 at both 80% (*v*/*v*) and 97% (*v*/*v*) concentrations.

#### 3.3.3. Antiviral Activity

AKV012 and AKV013 in 97.0% final concentration demonstrated antiviral activities (Figure 7) against the virus strains Adenovirus type 5 (ATCC VR-5), Poliovirus 1 Sabin strain, LSc-2ab (WHO, Geneva, Switzerland), and Murine Norovirus (Strain S99 Berlin) after a 60-s contact time in the presence of 0.3 g/L BSA at 20 °C.

For the antiviral activity, the product under test shall demonstrate at least a four-decimal log reduction in virus titre when tested in accordance with EN 14476+A1. These two AKVANO formulations thus demonstrated antiviral activity against the non-enveloped DNA adenovirus, the non-enveloped RNA poliovirus, and the non-enveloped RNA murine norovirus. According to the EN 14476 standard, products that have antiviral activity against these three virus strains are considered to be active against all other viruses. Considering the high ethanol concentration in the tested AKVANO formulations, the results are in line with earlier reports [19], although the additional benefit of citric acid is not obvious in the present study.

### 3.4. In Vitro Irritation Tests

#### 3.4.1. In Vitro Skin Irritation Test

Despite the potent antimicrobial activity, the AKVANO formulations are perceived as mild and non-irritating. To confirm this, two skin products based on AKVANO and intended to be used for skin care were tested according to the In Vitro EpiDerm™ Skin Irritation Test. According to the EU and GHS classification (R38/Category 2), an irritant is predicted if the mean relative tissue viability of three individual tissues exposed to the test substance is reduced below 50% of the mean viability of the negative controls. According to results obtained (Table 6) from this test, the viability of the reconstructed human epidermal model was >50%. This clearly shows that the AKVANO formulation (intended for use as a skin disinfectant) is classified as Non-Irritant (NI).

#### 3.4.2. In Vitro Eye Irritation Test

AKV011 is a prototype spray formulation intended to be used for treatment of plaque psoriasis, including affected areas on the scalp. Since spraying on the scalp implies a risk of exposing the eyes, the irritation potential of the formulation was tested.

The measured absorption values (blank subtracted) for the duplicate aliquots of each tissue included in the test were used to calculate viabilities for each tissue and mean viabilities for the test item and positive and negative controls together with the classification of the formulation to be evaluated AKV011 (Figure 8). If the viability is reduced to <50% of the negative control, the product is considered to have an irritating potential. Accordingly, AKV011 is considered not to have a potential for ocular irritation.

There was interference with the MTT testing, as determined by interference pretesting of the test substance, and freeze killed control tissues were used for AKV011 and negative control. Since the optical density values for freeze killed control tissues for the AKV011 sample were the same as for the negative control, no correction was needed.

### 3.5. In Vitro Permeation Experiments

Several AKVANO formulations containing the active pharmaceutical ingredients ketoprofen, diclofenac diethylamine, and diclofenac sodium were tested for permeation through Strat-M artificial membranes and compared to commercially available medicinal products.

The flux *J* (μg/h) was calculated according to Equation (3):(3)$$J_{i}=C_{i}V_{i}/t_{i}$$
where *i* is the fraction number, *C_i_* the concentration in μg/mL, *V_i_* the volume of the fraction (mL), and *t_i_* the time in hours during which the fraction was collected. The cumulative permeation *Q* was calculated according to Equation (4):(4)$$Q_{n}=\sum_{i=0}^{n}J_{i}t_{i}/m_{nom}$$
where *Q_n_* is the accumulated proportion of permeated active substance and *m_nom_* is the nominal amount of substance in µg applied to the membrane at the start of the experiment. Three different types of AKVANO vehicles were used in the study. AKV009a and AKV0010a contained only phospholipids, while AKV008, AKV009b, and AKV0010b also contained intermediate levels of single chain lipids MCM and IPM. AKV009c and AKV010c contained high concentrations of MCM and IPM, whereas another single chain lipid, oleic acid, was used in AKV009d at an intermediate level.

In the first set of experiments (Figure 9), a formulation of ketoprofen in AKVANO, AKV008, was compared to Orudis^®^ gel (2.5% ketoprofen, Sanofi AB, Stockholm, Sweden). The experiments demonstrated a much faster permeation profile for ketoprofen in AKV008 than for Orudis gel. Experiments also showed that a significant part of the initial content of ketoprofen in Orudis gel was retained on the membrane (47%), whereas for the AKV008 formulation, the retained amount was negligible (2.8%).

In a subsequent experiment, the permeation of diclofenac diethylamine in AKV009a-c was compared with Voltaren^®^ gel (2.3% diclofenac diethylamine, GlaxoSmithKline Consumer Healthcare ApS, Hovedstaden, Denmark). The result shows that the AKV009a formulation gives a comparatively slow release of diclofenac diethylamine, whereas AKV009b and AKV009c formulations, which contain increasing amounts of MCM and IPM, give faster permeation (Figure 10). AKV009c shows an even faster permeation than Voltaren gel, though the difference is not statistically significant (Figure 10). For all four formulations, a portion of initially applied diclofenac diethylamine was retained on the membrane (26% for AKV009a, 36% for AKV009b, 35% for AKV009c, and 26% for Voltaren gel) but the differences between formulations were not found to be statistically significant.

In another set of experiments, formulations of diclofenac sodium in AKVANO formulations AKV010a–d were tested. The trend is similar as for diclofenac diethylamine, though the permeation rate was generally slower (Figure 11). It is also observed that the amount retained on the membrane was higher for the AKV010a (30%) than for AKV010b (14%), AKV010c (17%) and AKV010d formulations (15%). The results from the experiments with diclofenac diethylamine and diclofenac sodium consistently show a higher permeation through the Strat-M^®^ membrane with increasing concentration of the single chain lipids MCM, IPM, and oleic acid in the formulation.

To sum up, the in vitro permeation data show that the AKVANO formulations can be designed to either enhance or to reduce the penetration of an incorporated active ingredient, simply by altering the lipid composition.

## 4. Conclusions

The presented novel drug delivery system for topical use, AKVANO, has been shown to possess advantageous features for formulation of pharmaceutical products as well as products for consumer health care and animal care. The properties can be tuned by changing the proportion between phospholipids and other lipids, such as single chain lipids. The volatile solvent system, based on ethanol or other short-chain alcohols, serves as an efficient solvent for lipids but also for a great number of active ingredients, and the phospholipids can also act as a solubilizer.

Investigations of AKVANO formulations’ in vitro characteristics, in terms of viscosity, aggregate size, diffusion coefficients, and physicochemical behaviour upon evaporation, show that the formulations are non-viscous, with virtually no or very minute aggregates formed. When formulations based on phosphatidylcholine are applied to the skin, e.g., by spraying a thin film consisting of lipid bilayer structures are formed. AKVANO formulations also meet the criteria for antibacterial, antifungal, and antiviral effects and, at the same time, can be classified as a non-irritant to the skin and eye. The in vitro skin permeation experiments on artificial skin mimicking membranes shows that a relatively slow permeation of the active ingredient can be obtained if only phospholipids are used. With increasing concentration of single chain lipids, such as medium chain monoglycerides and isopropyl myristate, the permeation can be increased significantly.

This first article about AKVANO formulations has thus presented the fundamental properties of the novel topical delivery system. With an understanding of the opportunities and limitations associated with AKVANO, it is possible to develop product prototypes with certain desired characteristics. Further development of pharmaceutical and consumer health products through AKVANO technology has led to additional non-clinical and clinical data which will be reported in future articles.

## 5. Patents

1. Carlsson, A.; Holmbäck, J. Lipid Layer Forming Composition for Administration onto a Surface of a Living Organism. International Patent Application WO 2011/056115 A8, 30 June 2011.

2. Herslöf; B.; Holmbäck, J. Topical Composition and Carrier for Administration of Pharmaceutical or Cosmetic Active Ingredients. International Patent Application WO 2014/178789 Al, 6 November 2014.

3. Herslöf, B.; Holmbäck, J. Sprayable Topical Carrier and Composition Comprising Phosphatidylcholine. International patent application WO 2015/072909 Al, 21 May 2015.

4. Herslöf, B.; Holmbäck, J. Topical Pharmaceutical, Cosmetic and Disinfectant Compositions Comprising Phosphatidylcholine. International Patent Application WO 2015/072910 Al, 21 May 2015.

## Figures and Tables

**Figure 1 pharmaceutics-14-00794-f001:**
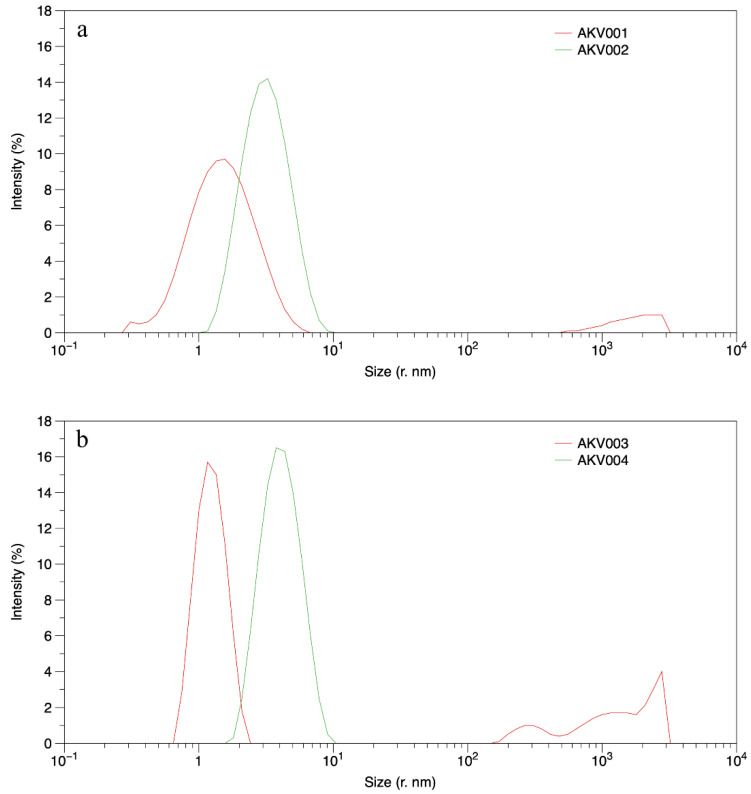
Average (triplicates) size distribution of AKVANO formulations based on DMPC (red) and DOPE (green) lipids, without (**a**) and with (**b**) additions of water, as measured by DLS measurements.

**Figure 2 pharmaceutics-14-00794-f002:**
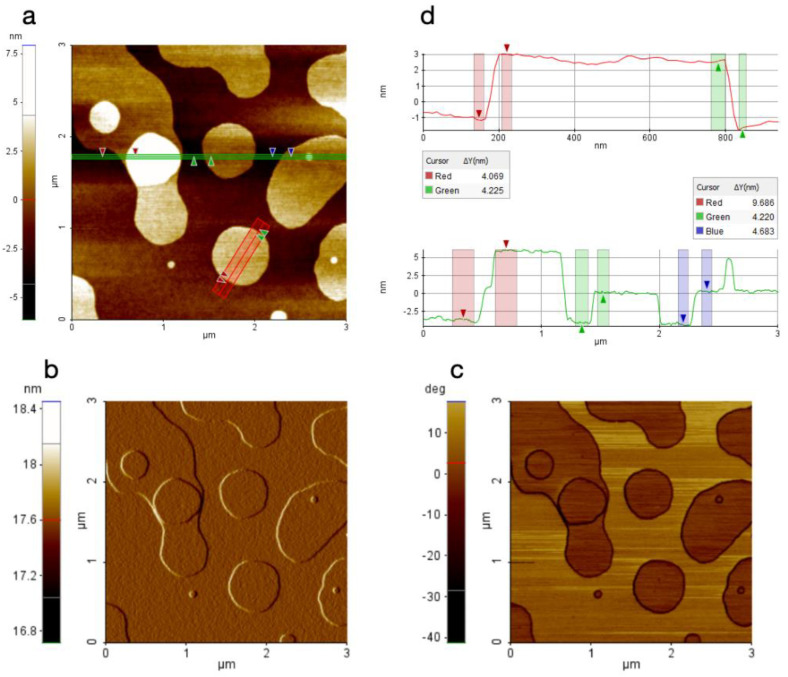
AFM image of AKV006 (DOPC) scanned at a location with a thin deposited lipid film, where (**a**) shows the topography data, (**b**) the amplitude signal, (**c**) the phase signal, and (**d**) the topography profile for the two lines drawn in the topography image.

**Figure 3 pharmaceutics-14-00794-f003:**
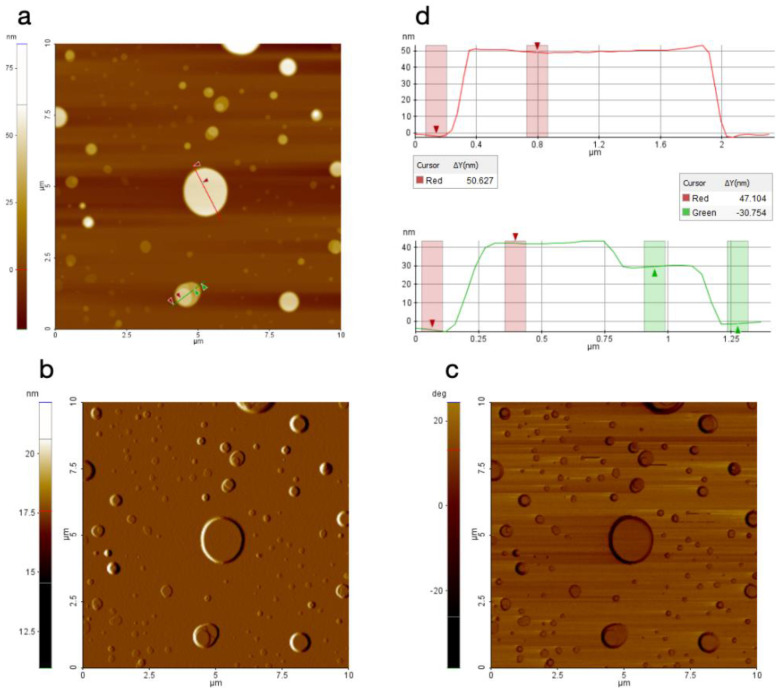
AFM image of AKV006 (DOPC) scanned at a location with “thicker” deposited lipid film, where (**a**) shows the topography data and (**b**) the amplitude signal (**c**) the phase signal, and (**d**) the topography profile for the two lines drawn in the topography image.

**Figure 4 pharmaceutics-14-00794-f004:**
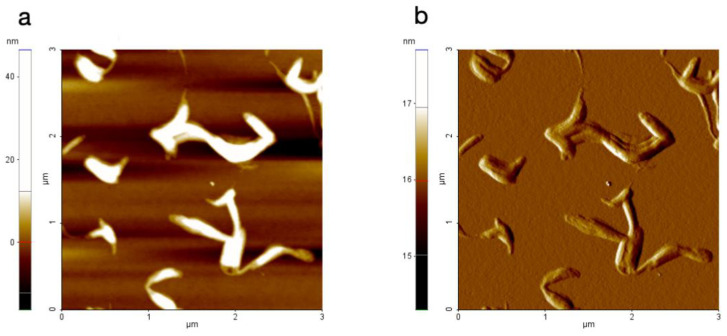
AFM image of AKV007 (DOPE) scanned at a location with a thin deposited lipid film, where (**a**) shows the topography data and (**b**) the amplitude signal.

**Figure 5 pharmaceutics-14-00794-f005:**
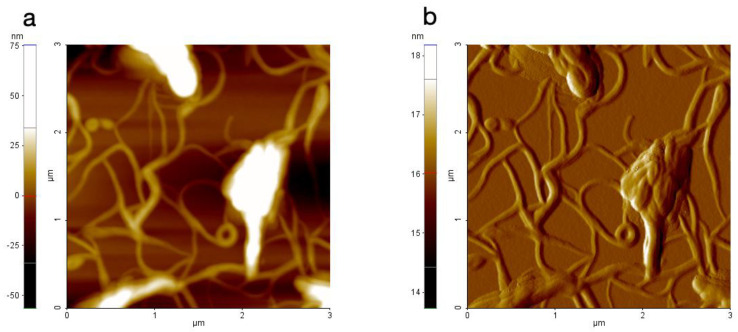
AFM image of AKV007 (DOPE) scanned at a location with “thicker” deposited lipid film, where (**a**) shows the topography data and (**b**) the amplitude signal.

**Figure 6 pharmaceutics-14-00794-f006:**
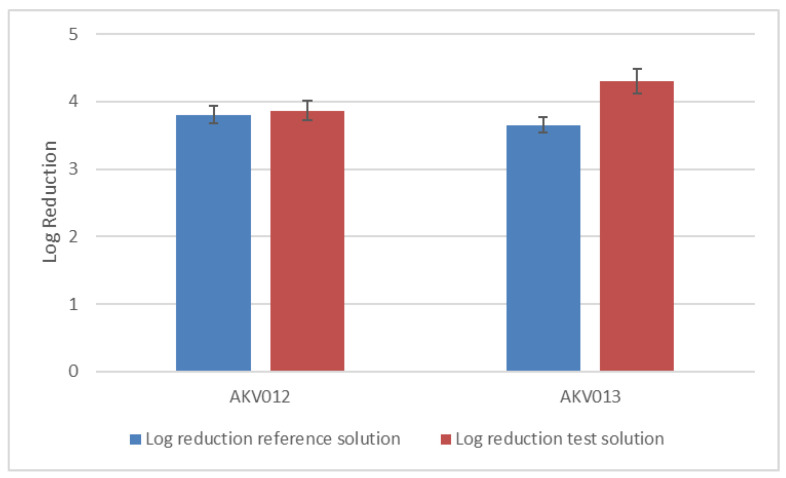
Log reduction ± SEM (standard error of mean) of AKVANO skin disinfectant sprays AKV012 and AKV013 compared to reference solution 2-propanol 60% (*v*/*v*).

**Figure 7 pharmaceutics-14-00794-f007:**
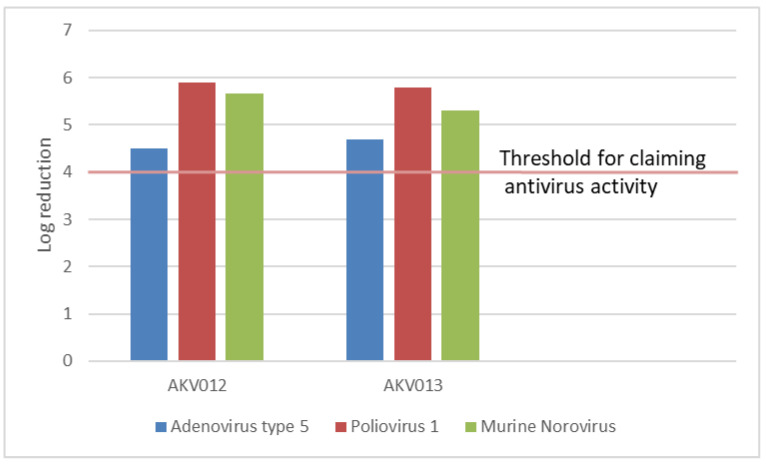
Log reduction of three virus strains by AKVANO skin disinfectant formulations AKV012 and AKV013. Calculations are based on the proportion of ten replicates showing cytopathic effects at a certain dilution level.

**Figure 8 pharmaceutics-14-00794-f008:**
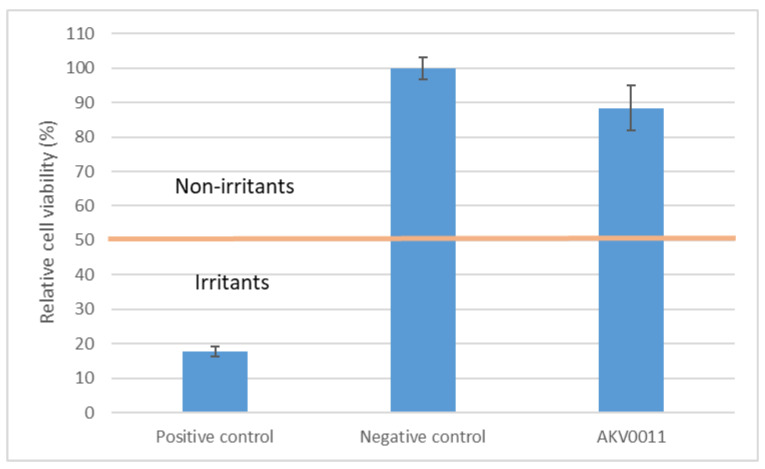
Viability of test item AKV011 in comparison with positive and negative controls after in vitro eye irritation test (percentage of negative control ± SEM).

**Figure 9 pharmaceutics-14-00794-f009:**
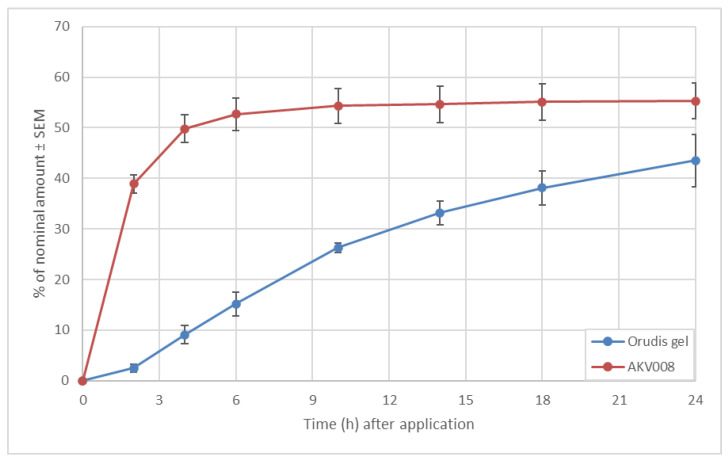
Cumulative permeation of ketoprofen through Strat-M membranes.

**Figure 10 pharmaceutics-14-00794-f010:**
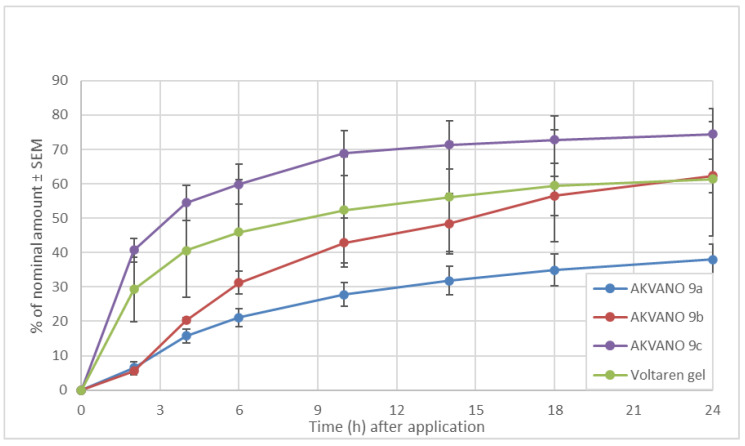
Cumulative permeation of diclofenac diethylamine through Strat-M membranes.

**Figure 11 pharmaceutics-14-00794-f011:**
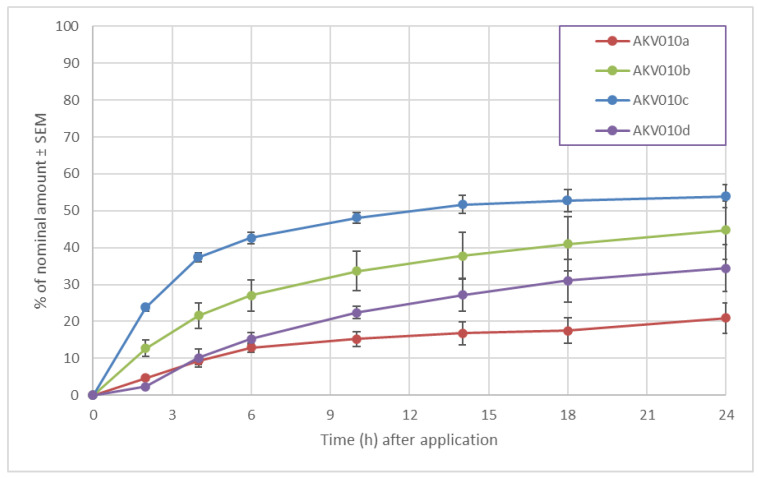
Cumulative permeation of diclofenac sodium through Strat-M membranes.

**Table 1 pharmaceutics-14-00794-t001:** Compositions used in the experiments.

Formulation	Phospholipid	% (*w*/*w*)	Alcohol	% (*w*/*w*)	Cyclomethicone % (*w*/*w*)	Active Agent	% (*w*/*w*)	Additional Components	% (*w*/*w*)
AKV001	DMPC	9.5	EtOH	23.2	67.3	-		-	
AKV002	DOPE	9.6	EtOH	23.1	67.3	-		-	
AKV003	DMPC	9.6	EtOH	23.1	65.8	-		Water	1.4
AKV004	DOPE	9.7	EtOH	23.0	65.8	-		Water	1.5
AKV005	-	-	EtOH	25.3	74.7	-		-	
AKV006	DMPC	10.1	EtOH	20.0	70.0	-		-	
AKV007	DOPE	9.9	EtOH	20.0	70.0	-		-	
AKV008	Soy-PC	17.9	EtOH	69.2	-	Ketoprofen	2.6	MCM IPM	5.0 5.3
AKV009a	Soy-PC	19.9	EtOH	77.8	-	Diclofenac diethylamine	2.3		
AKV009b	Soy-PC	9.6	EtOH	18.6	59.8	Diclofenac diethylamine	2.3	MCM IPM	4.8 4.9
AKV009c	Soy-PC	5.2	EtOH	72.5	-	Diclofenac diethylamine	1.9	MCM IPM	10.2 10.2
AKV010a	Soy-PC	21.1	EtOH	76.9	-	Diclofenac sodium	2.0		
AKV010b	Soy-PC	10.1	EtOH	77.1	-	Diclofenac sodium	2.1	MCM IPM	5.7 5.0
AKV010c	Soy-PC	4.9	EtOH	73.3	-	Diclofenac sodium	1.8	MCM IPM	9.9 10.1
AKV010d	Soy-PC	9.8	EtOH	78.1	-	Diclofenac sodium	2.1	Oleic acid	9.9
AKV011	Soy-PC	10.1	EtOH	19.7	60.1	Calcipotriol Betamethasone dipropionate	0.0056 0.053	MCM IPM α-tocopherol	4.9 5.0 0.12
AKV012	Soy-PL	5.2	Den. EtOH *	94.7	-	-		-	
AKV013	Soy-PL	5.2	Den. EtOH *	93.8	-	-		Citric acid	1.0
AKV014	Soy-Lec	8.1	Den. EtOH *	81.5		Lactic acid Urea Propylene glycol	2.7 2.5 5.1	Eucalyptus oil	0.1

* Denatured ethanol: Ethanol absolute 89%, 2-propanol 10%, and tert-butanol 1% (*w*/*w*).

**Table 2 pharmaceutics-14-00794-t002:** Dynamic viscosities of AKVANO formulations.

Formulation	Ingredients	Viscosity (mPa·s)
AKV001	DMPC	3.02
AKV002	DOPE	2.95
AKV003	DMPC + H_2_O	3.37
AKV004	DOPE + H_2_O	3.44
AKV005	Solvent	2.37

**Table 3 pharmaceutics-14-00794-t003:** Apparent average hydrodynamic radii (*R_h_*) with standard deviation and polydispersity index (PDI) obtained from the DLS experiments.

Formulation	Ingredients	*R_h_* (nm)	SD (nm)	PDI
AKV001	DMPC	1.7	0.11	0.14
AKV002	DOPE	3.4	0.15	0.12
AKV003	DMPC + H_2_O	1.4	0.20	0.27
AKV004	DOPE + H_2_O	4.3	0.02	0.09

SD—standard deviation.

**Table 4 pharmaceutics-14-00794-t004:** Self-diffusion coefficients of the different components in the respective formulation samples and apparent hydrodynamic radii (*R_h_*) of lipid aggregates as calculated from *D_lipid_*.

Formulation	Ingredients	*D_lipid_* (m^2^/s)	*D*_EtOH_ (m^2^/s)	*D*_D5_ (m^2^/s)	*D*_H2O_ (m^2^/s)	*R_h_* (nm) *
AKV001	DMPC	1.03 × 10^−10^	5.34 × 10^−10^	2.97 × 10^−10^	-	0.9
AKV002	DOPE	5.28 × 10^−11^	5.81 × 10^−10^	3.02 × 10^−10^	-	1.7
AKV003	DMPC + H_2_O	8.58 × 10^−11^	4.78 × 10^−10^	2.62 × 10^−10^	3.52 × 10^−10^	1.1
AKV004	DOPE + H_2_O	5.95 × 10^−11^	4.89 × 10^−10^	2.84 × 10^−10^	-	1.5
AKV005	Solvent	-	7.22 × 10^−10^	3.79 × 10^−10^	-	-

* Calculated from the Stokes-Einstein equation [5] using the value of the viscosity determined for the solvent mixture (AKV005), i.e., 2.37 mPa·s.

**Table 5 pharmaceutics-14-00794-t005:** Diffusion coefficients of the solvent components normalized to the diffusion coefficients in the pure solvent mixture in the absence of lipids, i.e., in sample AKV005.

Sample	(*D*/*D*_0_)_EtOH_	(*D*/*D*_0_)_D5_	1/Viscosity (mPa^−1^ s^−1^)
AKV001	0.74	0.78	0.33
AKV002	0.81	0.80	0.34
AKV003	0.66	0.69	0.30
AKV004	0.68	0.75	0.29

**Table 6 pharmaceutics-14-00794-t006:** Viability of reconstructed human epidermal after exposure to two AKVANO formulations and positive control, relative to the negative control.

Formulation	Relative Viability (%)	SD of Viability
Positive control	4.2	0.38
AKV012	71.5	7.99
AKV014	60.5	15.88

SD—standard deviation.

## Data Availability

The data presented in this study are available on request from the corresponding author. The data are not publicly available due to privacy issues.

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
