# Peer review of "AKVANO®: A Novel Lipid Formulation System for Topical Drug Delivery—In Vitro Studies"

_pharmaceutics, 2022, doi:10.3390/pharmaceutics14040794_

Round 1

Reviewer 1 Report

It is not clear or explained why only AKV001 to 5 formulations are analysed for DLS and NMR.
According to the authors "This is the first publication describing a novel topical delivery system called AKVANO". However, it is my understanding that the formulations selected should be exactly the same for all tests performed, both physicochemical, AFM, antimicrobial, irritation or skin permeation. In this way it is very difficult to correlate results. This is a major drawback in this paper.
DLS results are not properly presented. First, filtration through 0,45 um filter is not adequate, and is not justified. Results lack the pdi, or the sd for the claimed three measurements.
AFM results are not comparable. Authors have selected different formulations in different thicknesses to get images and measurements.
For antiviral activity, no statistic is presented. Either the author didn´t do replicates or the didn´t present the results

Author Response

Point 1: It is not clear or explained why only AKV001 to 5 formulations are analysed for DLS and NMR. According to the authors “This is the first publication describing a novel topical delivery system called AKVANO”. However, it is my understanding that the formulations selected should be exactly the same for all tests performed, both physicochemical, AFM, antimicrobial, irritation or skin permeation. In this way it is very difficult to correlate results. This is a major drawback in this paper.

Response 1: The purpose of the physiochemical characterization of some model systems was to understand the behavior of lipids when dissolved in the solvent system and after evaporation. This was part of the development of the technology and not meant to be used as method to characterize all feasible formulation examples. The conclusions made, e.g. selecting PC-rich lipids rather than PE-rich, could be expected to be rather universal. We will add a comment about this in the text.

Point 2: DLS results are not properly presented. First, filtration through 0,45 um filter is not adequate, and is not justified. Results lack the pdi, or the sd for the claimed three measurements.

Response 2: It is common to filter sample with 0.45µm filter prior to measurements to remove dust particles that would otherwise affect the accuracy of the measurements. There are no reasons to believe the the small particles that we are analyzing would be filtered off or get trapped in the filter. PDI and SD has been added to the table. The reason for filtering as well as comments regarding SD and PDI has been added in the text.

Point 3: AFM results are not comparable. Authors have selected different formulations in different thicknesses to get images and measurements.

Response 3: Two additional AFM figures have been added showing topography, amplitude, phase and profile for AKV006 imaged at the thick part, (Fig. 3) and topography and amplitude for AKV007 at thin part (Fig. 4). This will give a better comparison for both formulations imaged bot at thick and thin parts of the substrate.

Point 4: For antiviral activity, no statistic is presented. Either the author didn´t do replicates or they didn´t present the results.

Response 4: Ten replicates are used in the studies. However, the calculated log reduction of virus activity is based on the proportion of replicates showing CFE (cytopathic effects) at a certain dilution level. To our knowledge it is not possible to transform this to a measure of variation. This could be explained in the figure caption.

Reviewer 2 Report

This work developed a novel formulation technology called AKVANO® with the aim to provide a tuneable and versatile drug delivery system for topical administration. This drug delivery system is based on water-free lipid formulations where phospholipids and elective active ingredients are dissolved in volatile solvents. AKVANO formulations can be easily applied to the skin, e.g., by spraying, which provides a platform for topical delivery of various pharmaceutical ingredients. In summary, this article is recommended to be accepted with revisions in this journal.

  1. The stability of the AKVANO formulations should be tested.
  2. In Figure 9, The representative curves of AKV010c and AKV010d formulation group have the same color and cannot be distinguished. Please carefully check and revise Figure 9.

Author Response

Point 1: The stability of the AKVANO formulations should be tested.

Response 1: Some of the formulations used have been subjected to stability studies and this will be briefly summarized in the revised manuscript.

Point 2: In Figure 9, The representative curves of AKV010c and AKV010d formulation group have the same color and cannot be distinguished. Please carefully check and revise Figure 9.

Response 2: The colors will be adjusted in the revised manuscript.

Round 2

Reviewer 1 Report

Modifications made on the article together with authors answers to comments cleared out some doubts and improved the quality of the paper.